# Expanding the Toolbox for Functional Genomics in *Fonsecaea pedrosoi*: The Use of Split-Marker and Biolistic Transformation for Inactivation of Tryptophan Synthase (*trpB*) Gene

**DOI:** 10.3390/jof9020224

**Published:** 2023-02-08

**Authors:** Luísa Dan Favilla, Tatiana Sobianski Herman, Camila da Silva Goersch, Rosangela Vieira de Andrade, Maria Sueli Soares Felipe, Anamélia Lorenzetti Bocca, Larissa Fernandes

**Affiliations:** 1Laboratory of Applied Immunology, Institute of Biology, Campus Darcy Ribeiro, University of Brasília, Asa Norte, Federal District, Brasilia 70910-900, Brazil; 2Graduate Program in Molecular Biology, Institute of Biology, Campus Darcy Ribeiro, University of Brasília, Asa Norte, Federal District, Brasilia 70910-900, Brazil; 3Graduate Program in Molecular Patology, Faculty of Medicine, Campus Darcy Ribeiro, University of Brasília, Asa Norte, Federal District, Brasilia 70910-900, Brazil; 4Graduate Program in Microbial Biology, Institute of Biology, Campus Darcy Ribeiro, University of Brasília, Asa Norte, Federal District, Brasilia 70910-900, Brazil; 5Graduate Program of Genomic Sciences and Biotechnology, Catholic University of Brasilia, Campus Asa Norte, Asa Norte, Federal District, Taguatinga 70790-160, Brazil; 6Centro Metropolitano, Faculty of Ceilândia, Campus UnB Ceilândia, University of Brasília, Ceilândia Sul, Federal District, Brasilia 72220-275, Brazil

**Keywords:** gene replacement by homologous recombination, double-joint PCR, biolistic transformation, knockout mutant, *trpB*, tryptophan biosynthesis, *Fonsecaea*, chromoblastomycosis

## Abstract

Chromoblastomycosis (CBM) is a disease caused by several dematiaceous fungi from different genera, and *Fonsecaea* is the most common which has been clinically isolated. Genetic transformation methods have recently been described; however, molecular tools for the functional study of genes have been scarcely reported for those fungi. In this work, we demonstrated that gene deletion and generation of the null mutant by homologous recombination are achievable for *Fonsecaea pedrosoi* by the use of two approaches: use of double-joint PCR for cassette construction, followed by delivery of the split-marker by biolistic transformation. Through in silico analyses, we identified that *F. pedrosoi* presents the complete enzymatic apparatus required for tryptophan (trp) biosynthesis. The gene encoding a tryptophan synthase *trpB* —which converts chorismate to trp—was disrupted. The Δ*trpB* auxotrophic mutant can grow with external trp supply, but germination, viability of conidia, and radial growth are defective compared to the wild-type and reconstituted strains. The use of 5-FAA for selection of *trp*^-^ phenotypes and for counter-selection of strains carrying the *trp* gene was also demonstrated. The molecular tools for the functional study of genes, allied to the genetic information from genomic databases, significantly boost our understanding of the biology and pathogenicity of CBM causative agents.

## 1. Introduction

Chromoblastomycosis (CBM) is a chronic, progressive subcutaneous endemic mycosis prevalent in tropical and subtropical countries, caused by different black yeast-like fungi [1,2,3,4,5]. The predominant etiological agents change from one region to another; however, *Fonsecaea pedrosoi* is considered the most prevalent one worldwide [3,6,7,8,9,10,11]. After mechanical implantation of fungal saprophytic propagules into the subcutaneous tissue, the host develops CBM [1,2,4]. If lesions are undiagnosed and untreated, chronicity can be observed, leading to tissue fibrosis and lymphatic circulation impairment. In addition to the loss of limb functional capacity, malignant skin tumors and secondary infections can occur [1,2,5,12,13,14]. CBM is a difficult-to-treat disease with slow progression, challenging the usual antifungal therapies and leading to frequent relapses, as the causative agents do not respond well to the drugs. To bypass those limitations, alternative physical methods are required to remove the lesions [14].

For human pathogens, the capacity to infect and induce the disease depends on expression of virulence factors and secretion of metabolic products in host tissue. The best-described virulence factor in *Fonsecaea* species is melanin [15,16,17,18,19,20,21]. Several studies reported that this pigment plays a variety of roles in host–pathogen interactions, improving fungal survival, and promoting resistance to host immune response [22,23,24,25]. The mechanisms that *F. pedrosoi* elicits to infect and maintain the granulomatous inflammatory reaction in the host are still under investigation [26,27,28,29,30]. Inside infected tissue, this pathogen is able to differentiate into sclerotic cells, which are large, spherical, highly pigmented, thick-walled, and resistant to immune response [1,30,31,32,33]. The dimorphic transition is probably the main morphogenetic event required for CBM; the molecular mechanism and host factors that promote cell differentiation are not yet clear [1,34].

In recent years, our research group has worked to develop molecular tools for the functional study of genes in *F. pedrosoi*. We were the first to publish two genetic transformation systems in detail (biolistics and *Agrobacterium*-mediated transformation), as well as vectors carrying drug-resistance genes as markers of dominant selection for the construction of deletion cassettes [35]. Our advances have laid the foundation for the scientific community to explore the biology and the mechanisms of pathogenicity used by *F. pedrosoi*. In the present work, we took a step further in developing a useful and versatile molecular toolbox to perform genetic studies of *F. pedrosoi*.

Several organisms are capable of synthesizing tryptophan, including bacteria, plants, and fungi [36,37,38,39]. For all living organisms, tryptophan is not just an amino acid that builds proteins. It is also necessary for various physiological processes, serving as a substrate for secondary metabolite synthesis. Its role has recently been explored as a modulator of the immune response and maintenance of the human gut microbiota [40,41].

The enzymatic machinery for tryptophan anabolism is absent in animals and humans; however, they share a similar organization among the studied species so far [42,43,44,45,46,47]. Usually, it comprises four or five different enzymes in fungi, some of them with multifunctional characteristics, depending on the species. Chorismic acid from the shikimate pathway is the initial substrate to generate L-tryptophan [37,47,48]. The last reaction is catalyzed by tryptophan synthase enzyme [43,47,48]. In this work, we identified and in silico analyzed the putative enzymes for L-tryptophan biosynthesis by *F. pedrosoi*. Of the genes involved in the tryptophan biosynthetic pathway, none have been previously studied in *F. pedrosoi*; thus, we chose *trpB* (a putative tryptophan synthase gene) to generate a null mutant. In addition, we described in detail the construction of a deletion cassette in a quick and easy way (by double-joint PCR) [49] that can be directly delivered by biolistic transformation, without cloning steps, into conidia. Finally, we characterized the *trpB* null mutant phenotype and proposed new molecular tools for genetic manipulation of *F. pedrosoi*. Taken together, our data evidence that gene replacement by biolistic transformation is a feasible approach for functional genetic studies to advance the knowledge of this human fungal pathogen and other closely related species.

## 2. Materials and Methods

### 2.1. Strains and Growth Conditions of F. pedrosoi

*F. pedrosoi* CBS 271.37 was kindly provided by Dr. Vania A. Vicente from the University of Paraná and Paraná Network of Biological Collections. It was used in all experiments and maintained on Sabouraud dextrose agar (SAB) (for 1L: 10 g peptone, 40 g glucose, 15 g agar; pH: 5.6; autoclaved at 120 °C/15 min) at 28 °C for 7 days. Mutants carrying the *trpB* deletion were cultivated on the same media with the addition of 500 µg/mL of L-tryptophan (L-trp) (Sigma-Aldrich, St. Louis, MO, USA) from a 10 mg/mL stock solution prepared in ultrapure water, 0.45 μm filtered, and sterilized. For conidia purification*, F. pedrosoi* was grown in potato dextrose (PD) broth (Difco) for 10 days at 28 °C under 200 rpm of agitation. The culture was vortexed for 1 min, filtered through sterile glass wool, and centrifuged at 4000 rpm for 5 min. The pelleted conidia were resuspended in saline solution (0.9% NaCl), and the density was adjusted after counting in a Neubauer chamber. Potato dextrose agar (PDA) consisted of the PD broth with an addition of 1.5% of bacteriological agar. LB medium (1% peptone, 1% yeast extract, and 0.5% sodium chloride) was used for growing *Escherichia coli* DH5α in cloning experiments.

### 2.2. In Silico Search for Enzymes Related to Tryptophan Biosynthesis in F. pedrosoi and Evaluation of The trp Gene’s Expression

We analyzed the *F. pedrosoi* tryptophan anabolic pathway based on data provided by other studies of *Aspergillus fumigatus* [48], *Cryptococcus neoformans* [50], and *Saccharomyces cerevisiae* [47]. We used the BLASTp tool to search for *F. pedrosoi* putative enzyme sequences on EnsemblFungi (https://fungi.ensembl.org/index.html, accessed on 20 June 2020). We also calculated the percentage of identity between *A. fumigatus* and *F. pedrosoi* amino acid sequences using http://imed.med.ucm.es/Tools/sias.html (accessed on 20 June 2020). To identify the conserved domains of *F. pedrosoi* tryptophan biosynthesis putative enzymes, Pfam search was employed (https://pfam.xfam.org/search/sequence, accessed on 20 June 2020) and, based on KEGG (https://www.genome.jp/kegg/, accessed on 20 June 2020), the enzymatic nomenclature (EC) was applied. Multiple alignment of the sequences was performed by ClustalW (https://www.genome.jp/tools-bin/clustalw, accessed on 20 June 2020) to identify the connecting region between α and β catalytic domains of TrpB. We evaluated the expression of the genes after the growth of a WT strain on minimal medium (Czapeck dox-CD) [31] and CD+L-trp (500 µg/mL) as control by real-time PCR using the ΔΔ^CT^ method and *gapdh* (Glyceraldehyde-3-Phosphate Dehydrogenase, Z517_04938, from *F. pedrosoi* genomic database) as housekeeping normalizer. Briefly, the WT strain was grown in PD broth for 7 days at 28 °C under agitation (150 rpm). After this period, the fungal mass was collected, washed twice with saline solution, and inoculated in CD and CD+L-trp for 3 h at 28 °C/150 rpm. The fungal biomass was precipitated by centrifugation at 4000 rpm for 5 min and the pellet was washed twice with saline solution. The supernatant was discarded, the lysis solution (RLT) from the RNEasy^®^ Plant RNA Extraction Kit (Qiagen) and 100 µL of acid-washed glass beads (600 µm) were added to the pellet, which was subjected to lysis using the Precellys equipment (Bertin) in three cycles of 6500 rpm for 1 min intercalated by incubation on ice for 2 min. Then, the total RNA extraction followed as per the manufacturer’s instructions. cDNA was generated using a high-capacity cDNA kit (Thermo Fisher Scientific, Waltham, MA, USA) and real-time PCR was performed using SYBR Green kit Master Mix (Thermo Fisher Scientific) in a 7500 Fast Real-time PCR System. At least one primer of all targets spanned an exon–exon junction.

### 2.3. Construction of trpB Deletion Cassette by Double-Joint PCR (DJ-PCR)

The *trpB* deletion cassette was constructed using double-joint PCR (DJ-PCR), described by Kim and colleagues [49]. All oligonucleotides used in this work are described in Table 1. Firstly, regular PCR was used to amplify four fragments of the cassette: 5′ and 3′ flanking ORF regions of *trpB*, using, respectively, Fp128 + Fp129 (915 bp), Fp130 + Fp131 (947 bp), and genomic DNA from the wild-type (WT) strain (CBS 271.37) as template (Appendix A). The oligonucleotides Fp90 + Fp15 were used to amplify the 5′ fragment of the Hygromycin B (HygB)-resistance marker (Hyg^R^) and the combination Fp16 + Fp91 for 3′ Hyg^R^ using pSilent-1 [51] as the template for the reactions. The products 5′ and 3′ Hyg^R^ were 1109 bp and 987 bp, respectively (amplicons not shown in Appendix A). The second step consisted of two independent overlapping PCRs using a combination of previously obtained products: 5′ *trpB* flanking was fused to 5′ Hyg^R^ to produce the 5′ trpB + 5′ Hyg^R^ of 2024 bp (Appendix A). The other reaction yielded a product of 1934 bp corresponding to 3′ trpB + 3′ Hyg^R^ (Appendix A). The OneTaq 2 × Master Mix with Standard Buffer (NEB) was used for all amplifications, following the supplier’s recommendations. Once the two fragments were obtained, they were directly used for genetic transformation of *F. pedrosoi*.

### 2.4. Biolistics Transformation and Selection of trpB Mutant

The biolistics transformation was performed as previously described by [35]. Five micrograms of each fragment constructed above were precipitated on tungsten particles M10 (0.7 μm diameter, Biorad, Hercules, CA, USA). Next, 1 × 10^9^ conidia of *F. pedrosoi* were spread on SAB agar supplied with tryptophan (1000 µg/mL) and were genetically transformed using the Biolistics PDS-1000/He Particle Delivery System (Biorad), following the manufacturer’s instructions. Bombarded plates were wrapped with aluminum foil and incubated at 25 °C for 24 h. On the next day, cells were scraped and inoculated on SAB agar with L-trp (1000 μg/mL) and HygB (50 μg/mL) (Invitrogen, Waltham, MA, USA). After 21 days of incubation at 25 °C, the colonies were replicated on a series of four selective media plates: SAB agar, SAB agar with 50 μg/mL of HygB (SAB + HygB), SAB agar supplemented with 1000 μg/mL of L-trp (SAB + trp), and SAB agar containing both L-trp (1000 μg/mL) and HygB (50 μg/mL) (SAB + L-trp + HygB). To evaluate mitotic stability, WT and Δ*trpB* strains were cultured on the same medium used for the selection of transformants (SAB + L-trp + HygB) and incubated at 25 °C. Every seven days, the colonies that had grown in the non-selective plate (SAB + L-trp) were transferred to new plates of SAB + L-trp + HygB and SAB + L-trp. The procedure was repeated until four passages were completed. Hygromycin B and L-trp levels were 50 µg/mL and 1000 µg/mL, respectively. The transformants that were mitotically stable and were unable to grow in the absence of tryptophan were selected for genomic DNA extraction and PCR confirmation of *trpB* deletion.

### 2.5. PCR Confirmation of trpB Deletion

A PCR was used to confirm *trpB* gene deletion of chosen transformants. Reaction Fp120 + Fp121, using genomic DNA from the WT strain as template, was the positive control, the product of which is a 570 bp of *trpB.* The combinations Fp119 + Fp15 and Fp118 + Fp16 amplify, respectively, the 5′ region (2.5 kb for Δ*trpB* and no amplification for WT) and 3′ region of the *trpB* locus replaced by the Hyg^R^ cassette (2.4 kb for Δ*trpB* and no amplification for WT). The *trpB* full-length locus was accessed by Fp118 + Fp119 reaction (4.6 kb for Δ*trpB* and 5.1 kb for WT) (Appendix A). gDNA of *F. pedrosoi* WT was used as a control for the reactions.

### 2.6. Southern Blotting

Two transformants in which the *trpB* deletion was confirmed by the PCR and the WT strain of *F. pedrosoi* had their gDNA extracted. Twenty micrograms of gDNA were overnight restriction digested with StuI (3U/μg of gDNA) at 37 °C and electrophoresed in a 1.0% TAE 1 × agarose gel. The gel was blotted onto a charged nylon membrane (GE Healthcare) by capillary transfer. The probe was amplified with oligonucleotides Fp128 + Fp129 (corresponding to the 5′ flanking region of the *trpB* deletion cassette) producing a 915 bp fragment labeled with digoxigenin, following the supplier’s instructions (PCR DIG Probe Synthesis Kit, Roche, Basel, Switzerland). The hybridization, washing, and detection procedures were all done according to the manufacturer’s instructions (DIG Easy Hyb Granules, DIG Wash and Block Solution Set, Anti-Digoxigenin-AP, Fab fragments and CDP-Star ready-to-use, Roche). ImageQuant LAS 4000 (GE) equipment was used to read the signal. A 5.7 kb hybridization fragment was expected for Δ*trpB* and a 3.1 kb for the undeleted control.

### 2.7. Evaluation of Tryptophan Concentration Required for ΔtrpB Growth

The minimum concentration of L-tryptophan required to promote the *trpB* auxotrophic mutants’ growth was determined by incubating both WT and mutant strains (2.Δ*trpB*) on SAB agar supplemented with different concentrations of the amino acid (0, 20, 50, 100, 250, 500, and 1000 μg/mL), at 28 °C for 10 days. The radial growth of the colonies was used to compare the strains.

### 2.8. Radial Growth Measurement

To quantify colony growth, 2.∆*trpB,* 22.Δ*trpB + trpB,* 25.Δ*trpB + trpB*, and WT strains were point-inoculated with same amount of conidia (10^3^) onto the surface of SAB agar containing 500 μg/mL of L-trp (SAB + L-trp). The plates were incubated at 28 °C or 37 °C. Colonies’ diameters were measured every 3 days for 3 weeks. Two independent experiments were carried out in triplicate. Data collected were analyzed using the program GraphPad Prism. Mixed-effects analysis and Dunnett’s multiple comparisons test were used to compare variations between the control (WT) and test groups.

### 2.9. Evaluation of Conidia Germination

After growth on PD broth, conidia from 2.∆*trpB,* 22.Δ*trpB + trpB,* 25.Δ*trpB + trpB*, and WT strains were collected, counted on a Neubauer chamber, and then inoculated in 10 mL of SAB + L-trp broth (500 μg/mL of L-trp) at the concentration of 10⁶ conidia/mL. The cultures were incubated under continuous agitation (200 rpm) at 28 °C or 37 °C for 24 and 48 h. After incubation, the conidial suspension was vortexed and 100 µL were collected for microscopic inspection. One hundred randomly selected conidia from the samples of the same culture were counted to distinguish between germinated and non-germinated conidia. Conidia with a germ tube were considered germinated, and swollen ones were considered non-germinated. The experiment was performed in triplicate, repeated twice, and the data were analyzed using the program GraphPad Prism. Two-way ANOVA and Dunnett’s multiple comparisons test were used to compare variations between the control (WT) and test groups.

### 2.10. Conidial Viability Assay

For the viability assay, strains (2.*∆trpB,* 22.Δ*trpB + trpB,* 25.Δ*trpB + trpB*, and WT) previously grown on PD broth had the conidia collected and counted in a Neubauer chamber. Five hundred conidia of each strain were spread on three independent SAB + L-trp agar plates (500 μg/mL of L-trp), incubated at 28 or 37 °C for 10 days, and then the number of colonies was counted (CFU). The experiment was performed in triplicate and repeated twice. Data were analyzed using the program GraphPad Prism. Two-way ANOVA and Dunnett’s multiple comparisons test were used to compare variations between the control (WT) and test groups.

### 2.11. Determination of 5-Fluoroanthranilic Acid (5-FAA) Concentration for Selection of trp^-^ and Counter-Selection of trp^+^

Culture media were supplemented with 5-fluoroanthranilic acid (5-FAA) (Sigma-Aldrich) for tryptophan pathway counter-selection. The stock solution (100 mg/mL) was prepared in absolute ethanol. Since the concentration used for counter-selection of auxotrophic tryptophan mutants is very variable among different fungal species [52,53,54,55], we decided to test the interval from 0.25 to 3.0 mg/mL for *F. pedrosoi*. Conidia from the WT and 2.∆*trpB* strains were obtained as previously described, and density was adjusted. Five microliters of each serial dilution (1 × 10^7^ to 1 × 10^3^ /mL) were spotted on SAB agar containing 500 μg/mL L-trp and 5-FAA. The same procedure was done to evaluate the susceptibility of reconstituted strains to 5-FAA. Fungal growth was observed after 13 days of incubation at 25 °C. The experiment was repeated four times. In order to simulate the use of 5-FAA as agent to select *trp*- mutants, a high density of conidia (5 × 10^5^) of the WT and Δ*trpB* strains was spread on a set of conditions. The plates SAB agar, SAB + L-trp (500 µg/mL), SAB + L-trp + HygB (500 µg/mL of L-trp and 100 µg/mL of HygB), and SAB + L-trp + 5-FAA (500 µg/mL of L-trp and 0.5 mg/mL of 5-FAA) were incubated at 25 °C and inspected every 3 days until the 22nd day of growth.

### 2.12. Gene Replacement Strategy

Construction of plasmid pFpNAT + *trpB* for reconstitution of 2.Δ*trpB* was carried out as follows: First, the 5.1 kb fragment corresponding to the *trpB* gene was amplified from the WT genome with primers Fp119 and Fp118 (Appendix A). The product was subjected to double digestion by NdeI and SacI to generate a 4.7 kb fragment. This fragment was then cloned into the plasmid pFpNAT [35], previously opened with NdeI and SacI, using the enzyme T4 DNA ligase (Invitrogen) (Appendix A). The ligation was transformed into thermo-competent *E. coli* DH5α, and colonies that grew in LB medium containing 100 µg/mL of Ampicillin were selected for plasmid extraction. Cloning of the 4.7 kb fragment corresponding to the *trpB* gene was confirmed by NdeI / SacI digestion (Appendix A) and the plasmid was named pFPNAT+*trpB* (8.9 kb). The resulting flanking regions of *trpB* ORF were 923 and 1,263 bp for 5′ and 3′, respectively. Non-linearized pFPNAT+*trpB* was transformed on 2.Δ*trpB* conidia to obtain the reconstituted strains. The conditions for biolistic transformation were the same as those previously described for obtaining the *trpB* mutant [35,56]. To select *trp*^+^ transformants, SAB agar plates were supplied with 500 µg/mL of L- trp and 100 µg/mL of Nourseotricin (Nat). After 21 days of growth at 28 °C, colonies were transferred to SAB + Nat agar plates to detect the tryptophan prototrophic transformants, which were selected for gDNA extraction. PCR amplification with Fp120 and Fp121 that produced a 570 bp fragment of *trpB* was performed (Appendix A). We also tested whether the Hyg^R^ marker used for the gene deletion was preserved in the *trpB* locus after the transformation of pFpNAT+*trpB* by plating the colonies on SAB agar + 100 µg/mL Nat + 50 µg/mL HygB.

## 3. Results

### 3.1. Biosynthesis of Tryptophan in F. pedrosoi—An In Silico Analysis

By an in silico search for the enzyme sequences of the tryptophan biosynthesis pathway, we propose how this process may occur in *F. pedrosoi* (Figure 1). In the *F. pedrosoi* genome database, we found four genes encoding enzymes that work in the five-step tryptophan biosynthesis (*trpB*, *trpC*, *trpD*, and *trpE*) as shown in Table 1. By KEGG analysis, we observed well-conserved pathway components among the species *F. pedrosoi*, *S. cerevisiae* [37,47], *A. fumigatus* [48], and *C. neoformans* [50]. We observed a high identity percentage between the primary sequences of *F. pedrosoi* TrpB, TrpC, TrpD, and TrpE and those of *A. fumigatus* (>63%). In addition, using the Pfam search tool, we detected all conserved domains of each enzyme in the tryptophan pathway (Appendix A). We evaluated the expression of the genes after incubation of WT strain for 3 h on minimal medium in comparison to the minimal medium supplemented with 500 µg/mL of L-trp (control) by real-time PCR; however, we did not detect a genetic modulation of *trp* genes in response to the conditions analyzed (Appendix A**)**.

In all tryptophan-producing species, biosynthesis begins with chorismate, which is the product of the shikimate pathway. The first step is carried out by TrpE/TrpC, in which chorismate and glutamine are converted to anthranilate. Anthranilate, in turn—in the presence of phosphoribosyl phosphate—is converted to phosphoribosyl anthranilate by TrpD (Anthranilate phosphoribosyl transferase). In the third step, carboxyphenyl-amino deoxyribose phosphate synthesis occurs [37]. In *S. cerevisiae*, phosphoribosyl-anthranilate isomerase is encoded by TRP1 [47]. However, in *F. pedrosoi*, the TRP1 orthologue is not found, and this step is performed by Trp3^TrpC^, as is the case for *C. neoformans* and *A. fumigatus* [48,50]. TrpC is a trifunctional enzyme found in several other fungal species [37,57], and it participates in three steps of this pathway. In the *F. pedrosoi* TrpC sequence, we found the three catalytic domains required for indole 3-glycerolphosphate synthesis: N- terminal glutamine amidotransferase class-I (GATase), Indole-3-glycerol phosphate synthase (IGPS), and C- terminal phosphoribosyl-anthranilate isomerase (PRAI) (Appendix A). The last two stages of tryptophan biosynthesis are performed by tryptophan synthase (Trp5^TrpB^) [37,43,47]. In the *F. pedrosoi* TrpB putative sequence, we found two conserved signature domains: the tryptophan synthase α chain (positions 8–258 aa) and tryptophan synthase β chain pyridoxal-phosphate attachment site (positions 389–712 aa). Between those domains, there is a poorly conserved connector region of 83 aa, located between a residue of tyrosine (Y) and proline (P) that forms the channel necessary to couple the reactions (Appendix A).

### 3.2. Gene Disruption and Reconstitution in F. pedrosoi

To expand and make available a more versatile molecular toolbox for further genetic functional studies in *F. pedrosoi*, we chose the *trpB* gene to construct the deletion cassette. The disruption cassette was obtained using the approach described by [49] called double-joint PCR. It consists of successive PCR steps for fusing flanking fragments of the gene of interest to the 5′ and 3′ fragments related to the selective marker. Two products were obtained as expected (Appendix A), and were readily co-transformed into *F. pedrosoi* spores (strain CBS271.37) by biolistics according to the parameters we previously defined.

Three independent experiments were conducted. Firstly, from 49 transformants resistant to HygB, 4 were unable to grow in the absence of tryptophan (SAB). The second transformation generated 19 mutants Hyg^R^, and 2 mutants did not grow in SAB. These two mutants, named 1 and 2.*ΔtrpB*, were chosen for PCR and Southern blotting (Appendix A and Figure 2). In the third experiment, we recovered 146 colonies of Hyg^R^, and 9 mutants showed the tryptophan auxotrophic phenotype. Thus, we observed that the rate of homologous recombination for *trpB* in *F. pedrosoi* varied from 6 to 11%. All trp auxotrophic mutants exhibited mitotic stability after three passages in the non-selective medium (SAB + L-trp).

Gene replacement was confirmed initially by PCR using two combinations of oligonucleotides. For both auxotrophic tryptophan mutants (1 and 2.Δ*trpB*), we amplified the expected products (Appendix A). To confirm there was no ectopic integration of the deletion cassette on those two mutants’ genome, a Southern blot was performed. Through a *trpB*-specific probe (1 + 2 flanking region of the deletion cassette), we visualized the expected hybridization pattern after StuI restriction digestion of gDNA on the 2.Δ*trpB* mutant (5.7 kb) and WT (3.1 kb) strains. Despite 1.Δ*trpB* displaying the 5.7 kb fragment, a 6.0 kb hybridization fragment was also detected, indicating a genomic rearrangement after the Hyg^R^ insertion (Figure 2). After confirming 2.Δ*trpB* is a null mutant for the *trpB* allele, it was used in the following experiments. To ensure the observed tryptophan auxotrophic phenotype was due to *trpB* deletion, as a proof-of-concept, we reintroduced the *trpB* gene into the Δ*trpB* mutant genome (Appendix A). Of 360 recovered Nat-resistant transformants, 12 recovered the ability to produce tryptophan, and were designated Δ*trpB* + *trpB* reconstituted strains. Reinsertion of *trpB* was also confirmed by PCR, using oligonucleotides that amplified a 570 bp fragment of the gene (Appendix A). As we selected the phenotype trp^+^ Hyg^R^ Nat^R^ (Appendix A), we considered *trpB* insertion to have not occurred in the original locus. All the strains (2.∆*trpB,* 22.Δ*trpB + trpB,* 25.Δ*trpB + trpB*, and WT) used on the following experiments had the presence of selective markers and the auxotroph for tryptophan confirmed after growth in SAB agar supplemented with antibiotics (Nat and Hyg) and L-trp (500 µg/mL) (Figure 3).

### 3.3. External Supply of Tryptophan Is Required for Proper Growth of F. pedrosoi ΔtrpB

To define the lowest amount of tryptophan necessary to support the growth of the *F. pedrosoi* auxotrophic mutant, various concentrations were tested. The minimum L-trp required by different *trp* mutants from other fungi are very variable, from 10 µg/mL for *S. cerevisiae* [52] to 2000 µg/mL for *Nodulisporium* [58]. We evaluated growth of Δ*trpB* in SAB agar supplemented with 20, 50, 100, 250, 400, 500, and 1000 µg/mL of L-trp. After 10 days of incubation at 28 °C, we detected darkly pigmented colonies in all L-trp-supplied plates above 250 µg/mL. Thus, *F. pedrosoi* Δ*trpB* requires intermediate levels of L-trp to grow and synthesize melanin (Figure 4) in comparison to other fungi, but is unable to grow in the absence of an external source of this amino acid.

### 3.4. Deletion of trpB Causes Growth Delay and Viability Defects

To assess whether *trpB* deletion affects the *F. pedrosoi* life cycle, we measured the radial growth of colonies in SAB agar supplemented with 500 µg/mL of L-trp over 21 days at 28 and 37 °C. The mutant (2.*∆trpB)* showed a growth delay in relation to WT and reconstituted strains (22.Δ*trpB + trpB* and 25.Δ*trpB + trpB*) at 28 °C; however, at 37 °C this growth retardation was not so evident over the evaluated period (Figure 5A,B). Notably, the growth rate of all strains (WT, Δ*trpB*, and reconstituted strains) is slower at 37 °C than observed at 28 °C. We also determined if Δ*trpB* conidia had the ability to produce germ tubes and if they are viable (Figure 5C,D). After 24 h at 28 °C on SAB broth, about 25% of WT and reconstituted strains’ conidia were germinated, while Δ*trpB* failed to reach 20% of conidia germination. After 48 h of incubation at 28 °C, the germination defect remained for the mutant in comparison to the other strains (WT and 25.Δ*trpB + trpB),* but the percentage of germ tubes increased to close to 38% (Figure 5C,D). Differently from what we observed at 28 °C, we did not detect statistical differences in conidia germination among mutant and the other strains at 37 °C. Finally, we analyzed the ability of conidia to produce viable colonies (CFU) on SAB agar supplemented with 500 µg/mL tryptophan. As can be seen in Figure 5E, we detected a statistical difference in the number of colonies recovered on SAB agar when compared Δ*trpB* to the other strains at both temperatures tested. At 28 °C, an average of 103% of WT conidia resulted in viable colonies compared to 51% of Δ*trpB*, 86% of 22.Δ*trpB + trpB*, and 77% of 25.Δ*trpB + trpB.* The difference in viability rate of Δ*trpB* was also detected at 37 °C, in which only 41% of CFU were recovered for Δ*trpB*; meanwhile, the WT and reconstituted strains were in the range of 60% (Figure 5E). Taken together, our data indicate that Δ*trpB* has a radial growth delay and a reduced conidial germination rate at 28 °C. We also show that decreased viability of Δ*trpB* conidia is not a temperature-dependent phenotype. The reconstituted strains displayed phenotypes similar to the WT in the evaluated tests, indicating that the defects detected in the mutant are related to the absence of *trpB*.

### 3.5. trpB Deletion Induces 5-FAA Resistance in F. pedrosoi

In yeasts, mutants for tryptophan biosynthesis genes are resistant to 5-FAA [52,53,54,55]. This antimetabolite, when assimilated, is converted into the toxic product 5-fluorotryptophan. In our work, we decided to evaluate whether *trpB* deletion in *F. pedrosoi* confers resistance to 5-FAA, in order to expand the genetic manipulation tools for selection and counter-selection of mutants. For this purpose, we spotted serial dilutions of WT and *trpB* strains on SAB + L-trp plates supplied by increasing amounts of 5-FAA (0.25; 0.5; 1.0; 2.0; and 3.0 mg/mL). The *F. pedrosoi trpB* mutant has higher resistance to 5-FAA compared to the WT strain. After 13 days of incubation at 28 °C, we observed that the *trpB* mutant tolerates 0.5 mg/mL of 5-FAA at low conidial density (10^2^), while the WT grows only on 10^5^ and 10^4^ conidial spots. None of the strains grew on plates with >0.75 mg/mL of 5-FAA (Figure 6A). The reintroduction of *trpB* allele restored the Δ*trpB* susceptibility to the WT level (Figure 6B).

In order to analyze the use of 5-FAA as an agent to select tryptophan auxotrophic mutants, we plated WT and *trpB* mutant strains on different agar plates as follows: SAB agar, SAB + L-trp, SAB + L-trp + HygB, and SAB + L-trp + 5-FAA. Furthermore, to simulate a genetic transformation and evaluate the rate spontaneous mutants for *trp* auxotrophy, a high density of conidia (5 × 10^5^) was spread on the plates. As expected, in the SAB plates only the WT strain was able to grow, while in the SAB + L-trp + HygB condition only the Δ*trpB* strain grew, since resistance to HygB is guaranteed by the presence of the *hph* gene. Interestingly, in the condition of SAB + L-trp + 5-FAA, only *trp*^-^ colonies were observed and no WT colonies were detected on those plates (Appendix A). This experiment demonstrates that it is possible to use 0.5 mg/mL of 5-FAA as the sole selective agent for identification of tryptophan auxotrophic mutants, extending the range of applications related to *trpB* locus and establishing it as a potential genetic transformation marker. Taken together, our results indicate that the 5-FAA-supplemented medium can be an interesting approach for tryptophan auxotrophic mutant selection. In addition, 5-FAA can be applied as a counter-selector of transformants in the case of *trp*^-^ as a recipient strain in combination with the use of *trpB* as the selective marker in genetic transformation of *F. pedrosoi*.

## 4. Discussion

We first evaluated the enzymatic machinery in silico for tryptophan biosynthesis in *F. pedrosoi*. We identified that the enzymes in *F. pedrosoi* are displayed in a similar way to other filamentous fungi [48,50]. Unlike *S. cerevisiae*, *F. pedrosoi* has four of the five enzymes required to transform chorismate into tryptophan [47]. As reported for other ascomycetes and basidiomycetes, a single TRP1 ORF encoding a monofunctional PRA isomerase (PRAI) was not found in *F. pedrosoi*. Instead, it is fused to *trpC^TRP3^,* generating a trifunctional enzyme with the arrangement from N- to C- terminals of glutamine amidotransferase (GATase), Indole-3-glycerol phosphate synthase (IGPS), and PRAI [47,48,50,59,60]. Our analyses showed that *F. pedrosoi* encodes all catalytic domains required for the synthesis of tryptophan; this finding opens prospects and launches new tools for a detailed and functional exploration of this essential pathway for fungal survival. Interestingly, in *C. neoformans*, TRP3*^trpC^* and TRP5*^trpB^* are essential genes [50], but the same cannot be applied to other fungi in which auxotrophic mutants were previously isolated [53,58,61,62,63]. We detected the four transcripts of the trp biosynthetic pathway (*trpB*, *trpC*, *trpD*, and *trpE*) by RT-PCR after WT strain biomass was incubated for 3 h in minimal medium and minimal medium supplemented with L-trp. It was not possible to detect differential expression of these genes depending on the culture media and incubation time used; however, the mechanisms of regulation of the tryptophan pathway deserve to be elucidated in future work. Because *trp* genes are suitable as selective markers for genetic transformation in other fungi, associated to the fact that their knockout mutants are easily detected on genetic transformation plates, we decided to evaluate the DJ-PCR approach to construct the deletion cassette and the biolistic approach to deliver the DNA in *F. pedrosoi*. Then, as a proof of concept of the methodology, we functionally characterized the tryptophan synthase (*trpB*) putative gene in this human fungal pathogen.

Tryptophan synthase is conserved among bacteria, archaea, fungi, and plants, and is absent in mammals; however, only fungi present the two subunits fused into a single ORF in an α–β order [37,64]. For this reason, in those microorganisms, the enzyme is a homodimer, unlike in bacteria, where it is a tetrameric enzyme with a β–α organization [45]. The *F. pedrosoi* TrpB putative protein sequence presents both α and β catalytic domains: tryptophan synthase alpha chain, pyridoxal-phosphate-dependent enzyme, and the region that interconnects them [45]. FpTrpB possesses the longest connector region (83 aa) located between a residue of tyrosine (Y) and proline (P), described so far among fungal species (ranging from 40 to 69 aa) [44,63,64,65]. Definition of the three-dimensional structure of tryptophan synthases occurred after the completion of X-ray crystallography studies of the *S. thyphimurium* enzyme [66]. The connector region forms a channel required for the mechanical transfer of indole between the two catalytic sites, and in fungi, it is longer than in other organisms to allow proper folding of α and β subunits and, thus, catalyze their respective reactions [46,67]. The deletion of an 18 amino acid segment of this region in *S. cerevisiae* inactivates the enzyme, and reinsertion of an unrelated fragment restores its activity [67]. TrpB from *F. pedrosoi* shares all features already described for other fungi; however, a more detailed analysis of its three-dimensional structure may elucidate if a longer length of tunnel region promotes any effect in enzymatic catalytic properties.

In 2018, our group reported—through two efficient techniques, biolistic and *Agrobacterium* mediated transformation—the insertion and expression of exogenous DNA into the *F. pedrosoi* genome. We also explored three drug-resistance markers (NAT, HYG, and NEO) for mutant selection. Continuing our efforts to make available and improve new molecular tools for genetic manipulation of a CBM-causing agent, in this study, we developed a stable, fast, and efficient approach for achieving targeted gene disruption by biolistic transformation into *F. pedrosoi*. We utilized the Hyg^R^ marker to replace *trpB* and constructed a deletion cassette using the double-joint PCR technique. This technique has already been successfully employed for other fungi [49,68,69,70]; however, this work shows for the first time its use in constructing cassettes for *F. pedrosoi* gene disruption. DJ-PCR is based on only two successive rounds of DNA amplifications and does not require long and time-consuming vector construction steps, as is the case for *Agrobacterium*-mediated transformation in which the insertion of the deletion cassette into the T-DNA-carrying vector is required. Another advantage of constructing gene-disruption cassettes by DJ-PCR is that amplified products can be directly used in the genetic transformation of the fungus by electroporation or biolistics. Since homologous recombination inside the selective marker fragments is also required for mutant recovery, the chance of obtaining mitotically unstable transformants is reduced with the usage of the split-marker strategy, avoiding undesired genomic insertion [69]. Considering the non-homologous recombination mediated by the NHEJ pathway is more common in filamentous fungi than homologous recombination [71], this strategy contributes to generating a higher targeted-integration frequency. To our knowledge, this is the first report of the use of the DJ-PCR method followed by biolistic transformation to obtain null mutants in *F. pedrosoi*.

Three independent experiments yielded an efficiency of 6–11% triple-crossover recombination to promote the *trpB* replacement. Two other genes were successfully disrupted in our laboratory, with an HR rate of 2 and 5%. Our data are in accordance with those previously described for other filamentous fungi, in which a low efficiency of target integration into a specific locus is observed [71]. For other ascomycetes, this rate is quite variable: *A. fumigatus* has an HR efficiency of 5–10% [72,73], while in *N. crassa* it ranges from 2 to 23% [74,75]. Even within the same species, variable efficiency can be observed, as we have observed for *F. pedrosoi*. This fact can be attributed to the gene locus context, the size of the flanking region chosen to construct the deletion cassette, and the DNA delivery method [71]. For many fungi, especially ascomycetes, several former works inactivated the NHEJ machinery responsible for random integration of foreign DNA at ectopic genomic sites to circumvent this problem. The most common approach is to inactivate the ku70 or ku80 genes, creating NHEJ-deficient recipient strains that significantly improve the frequency of knockout mutants’ recovery [71,76]. To prevent ectopic integrations during complementation of null mutants from interfering with the virulence and other characteristics of the fungus, some authors have already reported genomic regions considered as safe for genetic integrations, such as the safe haven for *C. neoformans* [77,78] or *RPS10* locus in *Candida albicans* [79]. Although we have not yet evaluated a potential safe region for gene reconstitution in *Fonsecaea* spp., this tool will be of great relevance when it was described.

*trpB* is not an essential gene in *F. pedrosoi*; however, its absence directly affects its life cycle. Our results show that the *F. pedrosoi trpB* mutant is extremely dependent on an external source of tryptophan. Supplementation of culture medium with >250 µg/mL of this amino acid is required to sustain mycelial vegetative growth and sporulation, but 500 µg/mL (level of L-trp used in all phenotype tests) was not enough to maintain germination, viability and growth rate as in the WT strain. We did not assess whether the WT phenotypes of Δ*trpB* strain can be rescued at higher levels of L-trpB as reported for the *A. nidulans* [63]. In *A. nidulans* Δ*trpB*, although 10 µM (2 µg/mL) of tryptophan can trigger conidial germination and 100 µM (20 µg/mL) maintains the vegetative growth in solid medium, only higher L-trp concentrations (10 mM/2000 µg/mL) induce conidiospore production. However, even 50 mM (10,000 µg/mL) of tryptophan is not enough to generate viable ascospores and mature cleistothecia [63].

Yeast trp auxotrophic mutants require low levels of exogenous tryptophan [for *S. cerevisiae* 20 µg/mL (0.1 mM), *Hansenulla* 30 µg/mL (0.15 mM) and *Candida guilliermondii* 40 µg/mL (0.2 mM)] [52,53,55]. On the opposite way, some filamentous fungi, as well as *F. pedrosoi*, demand 10 to 100 times higher amounts of this amino acid [*Aspergillus niger* (1 mM), *Metarhizium anisopliae* (5 mM) and *Nodulosporium sp* (10 mM)] [58,61,62,80]. Eckert and colleagues suggested that the increased demand for tryptophan might be due to the limited ability to uptake the amino acid from the external source [63]. It is widely known that the anabolic and catabolic machinery of tryptophan, permeases and transporters, are finely tuned by intracellular levels of aromatic amino acids [48,50,64,81]. In filamentous fungal species, the transcriptional factor Cpc (cross pathway control) regulates the synthesis of aromatic amino acids, including tryptophan [59,63,82,83,84]. The high availability of exogenous tryptophan can also promote feedback repression of the initial steps of aromatic amino acids biosynthesis, and impair *trp^-^* mutants’ growth due to decreased synthesis of phenylalanine and tyrosine. To circumvent this defect, simultaneous tryptophan and phenylalanine supply improved the growth of *N. crassa trp* mutants [85]. Further investigation can clarify whether CpcA plays the same role in *F. pedrosoi* and whether the addition of external sources of aromatic amino acids would benefit mycelial growth. By the way, we detected, through an in silico BLASTP search, a sequence similar to CpcA in the genome of *F. pedrosoi* opening prospects for its future characterization.

Auxotrophic mutants and the anabolic genes that complement biosynthesis defects have been widely used as powerful molecular tools in the fungal genetic transformation system, since the detection of the prototrophic transformants is quick and easy and does not require high-cost antibiotics [76,86,87]. The use of auxotrophic mutants has also disadvantages to be consider. In *C. albicans URA3* auxotrophic mutant, the authors identified that ectopically integration of *URA3* gene affects different characteristics of the fungus, including the virulence [79]. The *trp* encoding enzymes have been applied as selective markers of several fungi [53,54,59,61,80,88] and in this work, we propose the use of *trpB* as a new selective marker in *F. pedrosoi*. We showed the complementation of the *F. pedrosoi* Δ*trpB* mutant with the wild type copy of gene, that in spite of had being inserted randomly into the genome, the mutants became prototrophic and recovery the WT traits.

We assessed the susceptibility of *F. pedrosoi* to 5-FAA. Our results showed this antimetabolite could be used as a marker of counter-selection in genetic transformation experiments using trp*^-^* as the recipient strain. This strategy is already widely used for yeasts [50,52,53,55]; however, as far as we know, there are no data for filamentous fungi. Due to the inability of the WT strain to grow on culture medium supplemented with 0.5 mg/mL of 5-FAA, it is a feasible and interesting tool for easily find *trp*^-^ mutants (positive selection) and also for identification of transformants complemented by *trpB* (counter selection) in *F. pedrosoi.* The counter selection methodology is indeed useful for recycling the selective marker for multiple gene deletion strategies on the same host strain, and it can be employed to select point mutations in genes related to the biosynthesis pathway [88]. Considering all this, the *trpB* mutant, despite some defective traits observed in conditions we tested, is not unfeasible as a recipient strain and can be used to generate other knockout strains; in addition, the *trpB* gene can be an alternative to the antibiotics as a selective marker for functional genomics in *F. pedrosoi.*

As tryptophan is an essential amino acid for fungal development and survival, and mammals acquire it from dietary sources, some authors suggest its biosynthesis apparatus may be an attractive selective therapeutic target [89]. In *C. neoformans*, an anthranilate synthase inhibitor (6-diazo-5-oxo-L-norleucine) was efficient in promoting in vitro cell death [50]. Although the role of *TRP3* and *TRP5* has not been evaluated in the pathogenicity of *C. neoformans* [50], in *A. fumigatus*, null mutants of *aroC* (chorismate mutase) and *trpA* play important roles in pulmonary and systemic aspergillosis infection models [90]. Considering that *F. pedrosoi* requires higher levels of tryptophan to survive (250 µg/mL) than that achieved in human serum (about 14 µg/mL) [91], and that mammals do not produce this amino acid, the search for specific inhibitors of tryptophan biosynthesis enzymes becomes a potential strategy to be explored against *Fonsecaea* spp. and other microbial pathogens [89,92,93].

## 5. Conclusions

In conclusion, to our knowledge, our work represents the first successful gene inactivation of the main CBM causative agent, *F. pedrosoi*. We demonstrate the use of split-markers for homologous integration allied to biolistic transformation were feasible to obtain Δ*trpB* mutants. The *trpB* gene can be exploited as an alternative and attractive selective marker for auxotrophic complementation of the *trpB* mutant. In addition, we showed the employment of 5-FAA as positive selection of *trp*^-^ mutants and as a new approach to facilitate identification of null mutants’ complementation in *F. pedrosoi*. Our data broaden the molecular toolbox by the use of traditional techniques for functional genome studies to improve the knowledge of not only *F. pedrosoi* biology but also of other CBM etiological agents. Notably, the functional studies of genes in the fungi of the genus *Fonsecaea* spp. are far behind compared to other fungal species, further contributing to the fact that many gaps in our knowledge of the disease remain, and CBM continues to be treated as a neglected tropical disease that affects many people around of the world.

## Figures and Tables

**Figure 1 jof-09-00224-f001:**
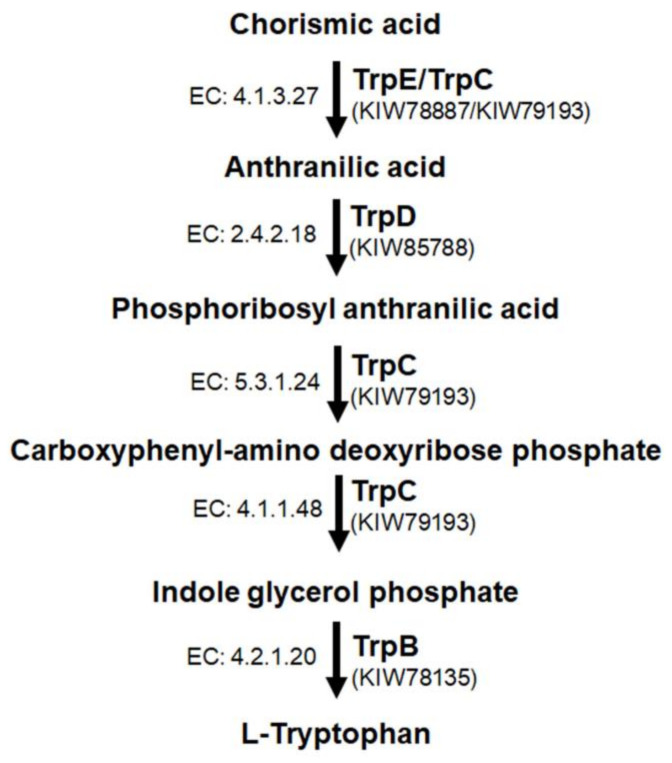
Putative tryptophan biosynthesis pathway in *F. pedrosoi.* The corresponding transcript IDs are indicated in parenthesis below their protein names. The *Enzyme Commission* (*EC*) numbers are depicted on the left side of each step of the pathway.

**Figure 2 jof-09-00224-f002:**
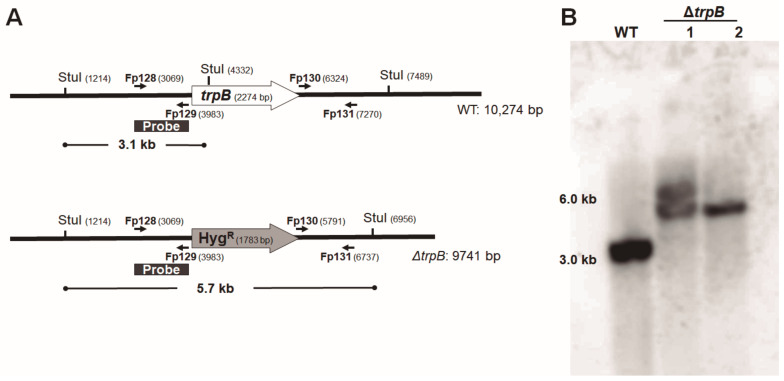
*trpB* deletion by homologous recombination of Hyg^R^ selective marker and confirmation by Southern blotting. (**A**) StuI restriction pattern of *trpB* locus in the genetic background of WT and mutant. (**B**) Southern blotting of the gDNA extracted from: WT, 1.∆*trpB*, and 2.∆*trpB* digested with restriction enzyme StuI probed with a 914 bp DIG-labeled PCR fragment. The annealing region of the probe is indicated as a solid line. The indications 1.∆*trpB* and 2.∆*trpB* correspond to two different null mutants for *trpB,* while the WT is the CBS 271.37.

**Figure 3 jof-09-00224-f003:**
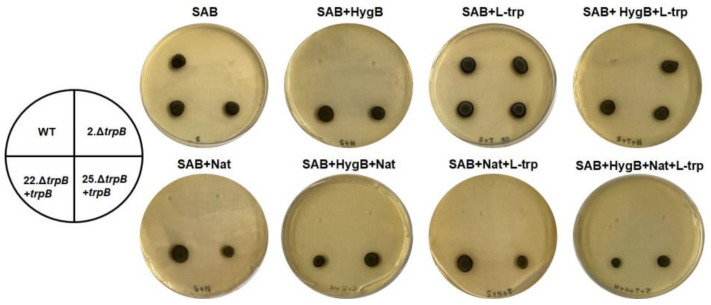
Radial growth of *F. pedrosoi* tryptophan auxotrophic mutant (2.Δ*trpB)*, reconstituted transformants (22.Δ*trpB + trpB* and 25.Δ*trpB + trpB)*, and WT on Sabouraud agar (SAB) supplemented or not with 500 µg/mL L-tryptophan (L-trp) and 50 µg/mL Hygromycin B (HygB). The 2.∆*trpB* corresponds to the null mutant for *trpB,* while 22.Δ*trpB + trpB* and 25.Δ*trpB + trpB* are two independent reconstituted strains for *trpB* and the WT is the CBS 271.37. The plates were incubated at 28 °C for 7 days.

**Figure 4 jof-09-00224-f004:**
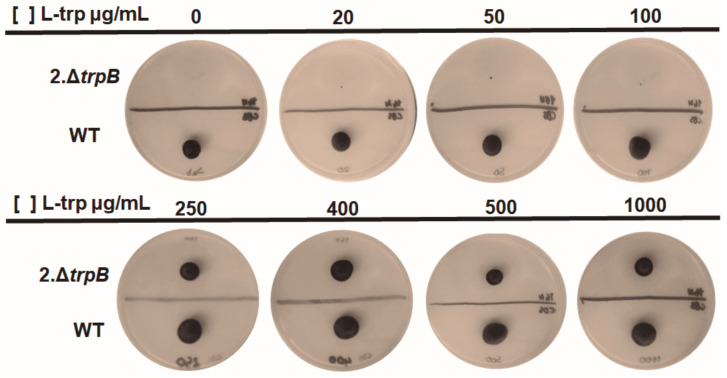
Determination of the minimum tryptophan level required for growth of *F. pedrosoi trpB* mutant. The 2.Δ*trpB* and WT strains were inoculated on SAB agar, supplemented or not with different amounts of tryptophan (L-trp) (0, 20, 50, 100, 250, 400, 500, and 1000 µg/mL). The WT is the CBS 271.37. The plates were incubated at 28 °C for 10 days.

**Figure 5 jof-09-00224-f005:**
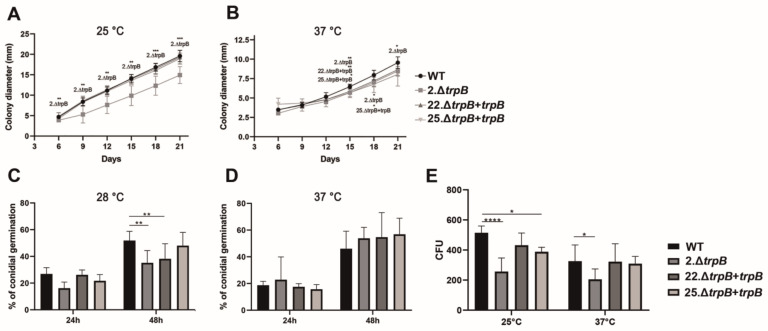
Deletion of *trpB* affects growth, viability, and conidial germination of *F. pedrosoi*. Colony diameter of WT, Δ*trpB*, and reconstituted strains was measured after inoculation on SAB agar. The radial growth of the colonies was measured every seven days at 25 (**A**) and 37 °C (**B**). The statistical analyses were mixed-effects analysis and Dunnett’s multiple comparisons post-test. *p* < 0.0332, *p* < 0.0021, and *p* < 0.0002 are indicated as *, **, and ***, respectively. Percentage of germinated conidia of WT, Δ*trpB*, and reconstituted strains at 28 (**C**) and 37 °C (**D**) after incubation on SAB+L-trp broth for 24 and 48 h. (**E**) Colony Forming Units (CFU) were counted after 500 conidia of each strain were plated on SAB L-trp and incubated for 10 days at 28 or 37 °C. The statistical tests were two-way ANOVA and Dunnett’s multiple comparisons post-test; **** indicates *p* < 0.0001. All experiments used culture medium supplemented with L-tryptophan (500 µg/mL). Averages of three independent experiments with intra-experiment triplicates were plotted on the graphics. The bars represent the standard error of each group.

**Figure 6 jof-09-00224-f006:**
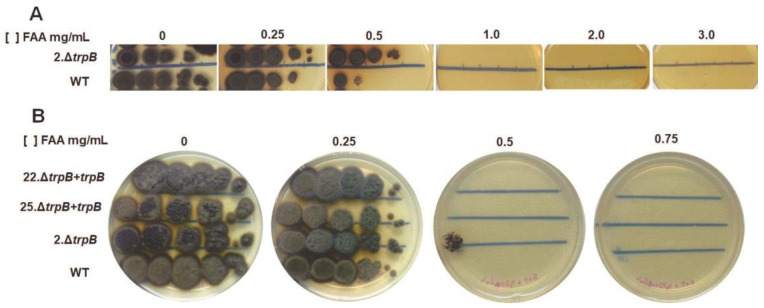
Deletion of *F. pedrosoi trpB* induces resistance to 5-FAA. The strains were grown on PD broth supplemented with 500 µg /mL of tryptophan (L-trp) for 7 days at 28 °C. The conidia were purified, counted, and serial diluted to 2 × 10^8^ − 10^3^/mL. Five µL of each dilution were spotted on SAB + 500 µg /mL of L-trp, supplemented with increasing concentrations of 5-FAA. (**A**) Serial dilutions of 2.∆*trpB* and the WT (CBS 271.37) conidia were spotted and plates were incubated at 28 °C for 13 days. The *trpB* mutant tolerates 0.5 mg/mL of 5-FAA at low conidia density (10^2^), while WT grows only on 10^5^ and 10^4^ spots. None of the strains were able to grow at 5-FAA concentrations above 0.75 mg/mL. (**B**) Susceptibility to 5-FAA requires a functional tryptophan biosynthesis pathway. Reconstituted transformants (22.∆*trpB + trpB* and 25.∆*trpB + trpB*) recovered the sensitivity to 5-FAA of ∆*trpB* to WT levels after incubation for 7 days at 28 °C. The pictures are representative of three independent experiments.

**Table 1 jof-09-00224-t001:** *F. pedrosoi* tryptophan biosynthesis genes and putative protein function in comparison to other fungi.

Enzymatic Function	*Protein Name /Gene ID*				*F. pedrosoi* transcript ID ^e^	% Amino acid identity ^f^
	*S. cerevisiae* ^a^	*C. neoformans* ^b^	*A. fumigatus* ^c^	*F. pedrosoi* ^d^		
Anthranilate synthase component I-Chorismate aminase [EC:4.1.3.27]	Trp2 YER090W	Trp2 CNAG_06679	TrpE Afu6g12580	TrpEZ517_08726	KIW78887	70.98
Anthranilate synthase component II-Glutamine amidotransferase/Phosphoribosyl-anthranilate isomerase/Indoleglycerol phosphate synthase [EC:4.1.3.27 4.1.1.48]	Trp3 YKL211C	Trp3 CNAG_04501	TrpC Afu1g13090	TrpCZ517_05805	-KIW79193	66.49
Anthranilate phosphoribosyl transferase [EC:2.4.2.18]	Trp4 YDR354W	Trp4 CNAG_00811	TrpD Afu4g11980	TrpDZ517_01180	-KIW85788	63.84
Phosphoribosylanthranilate isomerase [EC:5.3.1.24]	Trp1 YDR007W	Trp3 CNAG_04501	TrpC Afu1g13090	TrpCZ517_05805	-KIW79193	66.49
Tryptophan synthase [EC:4.2.1.20]	Trp5 YGL026C	Trp5 CNAG_00649	TrpB Afu2g13250	TrpBZ517_07968	-KIW78135	78.39

^a^ Sequence ID from *S. cerevisiae* genome database: https://www.yeastgenome.org accessed on 20 June 2020; ^b^ sequence ID from *C. neoformans* H99 strain: https://fungi.ensembl.org accessed on 20 June 2020; ^c^ sequence ID from *A. fumigatus* Af293 strain: http://www.aspergillusgenome.org accessed on 20 June 2020; ^d^ sequence ID from *F. pedrosoi* CBS271.37 strain https://fungi.ensembl.org accessed on 20 June 2020; ^e^ transcript ID from *F. pedrosoi* CBS271.37 strain https://fungi.ensembl.org accessed on 20 June 2020; ^f^ percentage of identity between *A. fumigatus* and *F. pedrosoi* protein sequences: pairwise alignments were done in https://www.ebi.ac.uk accessed on 20 June 2020 and identity calculation was performed at http://imed.med.ucm.es/Tools/sias.html accessed on 20 June 2020.

## Data Availability

Not applicable.

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
