# Peer review of "Expanding the Toolbox for Functional Genomics in Fonsecaea pedrosoi: The Use of Split-Marker and Biolistic Transformation for Inactivation of Tryptophan Synthase (trpB) Gene"

_jof, 2023, doi:10.3390/jof9020224_

Round 1

Reviewer 1 Report

In the submitted manuscript by Favilla et al., entitled “Expanding the toolbox for functional genomics in Fonsecaea pedrosoi: the use of split-marker and biolistic transformation for inactivation of tryptophan synthase (trpB) gene” the authors demonstrated the deletion of trpB gene in Fonsecaea pedrosoi. The authors utilized double joint PCR for cassette construction and subsequent biolistic transformation to generate the trpB mutant.

Overall, this study provides an important tool to expand the genetic information of this human pathogen. Manuscript writing is understandable and fairly easy to read. However, as I detail below, there are a few issues that must be addressed.

How efficient is the biolistic transformation compared to other transformation methods that are used? It would be better to show and discuss how this method is more efficient and faster compared to other transformation methods.

Fig. 5 Why did you use two different temperatures to test growth, viability, and conidial germination? Results indicate that trpB doesn’t affect these phenotypes at 37°C (human temperature). What’s the relevance of this data in establishing trpB gene as an alternative selective marker? Please explain.

Reviewer 2 Report

The manuscript by Favilla et al discussed the newly developed split-marker and biolistic transformation for functional genomics in F. pedrosoi

Manuscripts is well written with a clear structure. 

I only have few comments:

Line 433-434: Does the author mean WT grows on the lowest dilution? WT also grow a bit on the second lowest dilution conidial spots.

In figure 4, it seems like 2.∆trpB mutant grow slower in 500 ug/ml L-trp when compared to 400 ug/ml L-trp but grow faster in 500 ug/ml L-trp when compared to 1000 ug/ml?
